Article 

# Uncovering a miltiradiene biosynthetic gene cluster in the Lamiaceae reveals a dynamic evolutionary trajectory

Abigail E. Bryson [1,7], Emily R. Lanier [1,7], Kin H. Lau[2,3], John P. Hamilton[2,4], Brieanne Vaillancourt [2,4], Davis Mathieu[1], Alan E. Yocca [2,5], Garret P. Miller [1], Patrick P. Edger[5], C. Robin Buell[2,6] & Björn Hamberger [1] ✉

The spatial organization of genes within plant genomes can drive evolution of specialized metabolic pathways. Terpenoids are important specialized metabolites in plants with diverse adaptive functions that enable environmental interactions. Here, we report the genome assemblies of *Prunella vulgaris*, *Plectranthus barbatus*, and *Leonotis leonurus*. We investigate the origin and subsequent evolution of a diterpenoid biosynthetic gene cluster (BGC) together with other seven species within the Lamiaceae (mint) family. Based on core genes found in the BGCs of all species examined across the Lamiaceae, we predict a simplified version of this cluster evolved in an early Lamiaceae ancestor. The current composition of the extant BGCs highlights the dynamic nature of its evolution. We elucidate the terpene backbones generated by the *Callicarpa americana* BGC enzymes, including miltiradiene and the terpene (+)-kaurene, and show oxidization activities of BGC cytochrome P450s. Our work reveals the fluid nature of BGC assembly and the importance of genome structure in contributing to the origin of metabolites.

Plants are renowned for their incredible diversity of specialized metabolites, which function in interactions with their environment. These biosynthetic pathways are dynamic, facilitating continual evolution of novel compounds. The rising number of high-quality plant genomes published in recent years has led to the discovery that some metabolic pathways are organized into biosynthetic gene clusters (BGCs). A BGC is a group of two or more different classes of non-homologous genes which are physically clustered, transcriptionally linked, and functionally related[1–6]. Over 30 plant BGCs have been functionally validated to date[7] since the discovery of the first BGC in maize[8]. The BGCs found in plants are predominantly involved in specialized rather than central metabolism[9] and occur in multiple classes of compounds including benzylisoquinoline alkaloids in poppy[10,11], triterpenoid cucurbitacins in Cucurbitaceae[12,13], and diterpenoid momilactones in Poaceae and other cereals[14–18].

How and why BGCs form is still a topic of discussion, although several hypotheses are emerging. In bacteria and fungi, BGCs are common and aid in transference of the entire pathway during horizontal gene transfer[19,20]. While there is no evidence of horizontal gene transfer of plant BGCs reported thus far, BGCs still offer advantages in vertical inheritance of biosynthetic pathways[5,21]. The genetic linkage conveyed by BGCs facilitates coinheritance, which can protect the integrity of the entire pathway[22–24]. In some pathways, such as momilactone biosynthesis, loss of a single gene would result in a buildup of toxic intermediates[23]. Another fitness benefit of BGCs is the possibility of coregulation, such as by a single transcription factor or regulatory region. This can provide an energetically favorable control of the

[1]Department of Biochemistry, Michigan State University, East Lansing, MI, USA. [2]Department of Plant Biology, Michigan State University, East Lansing, MI, USA. [3]Bioinformatics and Biostatistics Core, Van Andel Institute, Grand Rapids, MI, USA. [4]Center for Applied Genetic Technologies, University of Georgia, Athens, GA, USA. [5]Department of Horticulture, Michigan State University, East Lansing, MI, USA. [6]Plant Resilience Institute, Michigan State University, East Lansing, MI, USA. [7]These authors contributed equally: Abigail E. Bryson, Emily R. Lanier. ✉e-mail: hamberge@msu.edu

metabolite production in a tissue or developmental stage-specific manner[5,16,21,25–28]. Regulation may also take place at the chromatin level, with DNA and histone methylation regulating transcription of the entire cluster[25,29–31].

Since the study of plant BGCs is still in its infancy, their origins and evolution are also not well understood. So far, evidence supports that plant BGCs have likely arisen from gene or genome duplication and/or genomic rearrangements[5]. BGC formation may be enhanced in highly active regions of the genome, such as the recent work detailing assembly of the oat avenacin BGC in a sub-telomeric region[32]. The birth of a gene cluster may begin with a single colocalized gene pair. Subsequent colocalization of additional classes of enzymes can occur through chromosomal remodeling or transposition[5,21,30,33]. Expansion of the cluster can also continue through tandem, local, or whole genome duplication[4,6,33–35]. The inherent promiscuity of enzymes involved in specialized metabolism enables rapid neofunctionalization, promoting functional divergence of BGCs as they evolve through different plant lineages[34,36–38]. Recent work has shown the conservation of core genes and diversification into new functions/pathways when comparing BGCs across different plant families[6,39].

Terpenoids are a class of specialized metabolites that are well represented among the studied BGCs. Plant terpenoids are incredibly diverse and encompass over 65,000 structures[40], making them the largest known class of plant natural products. Plants rely on terpenoids for many interactions, including pathogen and herbivore defense, signaling, and pollinator attraction[41–43]. Terpene synthases (TPSs) catalyze the formation of terpene backbones from diphosphate isoprenoid precursors and are classified into eight subfamilies (a-h) based on their phylogenetic relationships[41,44,45]. The bicyclic labdane-type diterpenes are typically formed by the sequential activity of a class II (TPS-c) followed by a class I (TPS-e) diterpene synthase (diTPS). Class II diTPSs catalyze a proton-mediated cyclization of a 20-carbon isoprenoid diphosphate, usually geranylgeranyl diphosphate (GGPP), to form the characteristic decalin core. A class I diTPS then cleaves the diphosphate and may further differentiate the diterpene backbone. Diterpene backbones are functionalized by other enzyme classes through oxidation and subsequent conjugation to increase bioactivity. Cytochromes P450 (CYPs), particularly in the expansive CYP71 clan, often oxidize terpenes and have been found colocalized with TPSs either as pairs or as expanded BGCs[2,46].

Terpenoid diversity is particularly rich in the Lamiaceae (mint) family[47,48]. Genome assemblies for 22 different Lamiaceae species (Supplementary Table 1) have been published to date, revealing BGCs for at least two classes of terpenoids: monoterpene-derived nepetalactones from catnip (*Nepeta* sp.)[49] and diterpenoid tanshinones in the Chinese medicinal herb Danshen (*Salvia miltiorrhiza*)[24,50,51]. Tanshinones are studied for their potent pharmacological activities, and as a result much of the biosynthetic pathway has been elucidated (Supplementary Fig. 1)[24,50–59]. The terpene backbone of the tanshinones is miltiradiene, a labdane diterpene formed by a class II (+)-copalyl diphosphate ((+)-CPP) synthase followed by the class I miltiradiene synthase. The abietane-type diterpenoid miltiradiene is the likely terpene precursor to a wide array of bioactive diterpenoids that are common throughout the Lamiaceae and beyond[60]. The antimicrobial effects demonstrated for many of these terpenoids suggest a native role in plant defense[60–64]. Carnosic acid is another abietane diterpenoid found in several Lamiaceae species with powerful antioxidant and anticancer properties[65]. The biosynthesis of carnosic acid and related diterpenoids has been elucidated in *Rosmarinus officinalis, Salvia pomifera* and *Salvia fruticosa* (rosemary and sages)[66,67] and involves many CYPs orthologous to those involved in tanshinone biosynthesis (Supplementary Fig. 1).

Previous studies of the *S. miltiorrhiza* genome have found two BGCs that together contain the genes encoding miltiradiene diTPSs and two CYP76AHs involved in tanshinone biosynthesis[24,50,51]. A third

locus containing an array of CYP71Ds includes the two genes for the enzymes (CYP71D375 and CYP71D373) responsible for the D-ring heterocycle of the tanshinones. Recent publication of additional Lamiaceae genomes revealed syntenic BGCs in four other species: *Tectona grandis, Salvia splendens* and *Scutellaria baicalensis* (teak, scarlet sage and Chinese skullcap, respectively)[24,58,68]. Additionally, we previously reported the presence of a large cluster in *Callicarpa americana* (American beautyberry) which contains orthologs of the miltiradiene diTPS genes as well as those encoding multiple CYP76AHs and CYP71Ds[69]. The divergence of these five species indicates that this BGC may be present ubiquitously throughout the Lamiaceae.

In this work, to explore the prevalence and evolution of the miltiradiene BGC, we survey a representative panel of 10 Lamiaceae genome assemblies (Fig. 1). We focus on synteny with the BGC in *C. americana*, which is one of the largest yet discovered, spanning approximately 400 Kb and encompassing seven diTPSs and twelve CYPs. Our syntenic analysis shows the conservation of core miltiradiene biosynthetic genes throughout all species studied while highlighting lineage-specific diversification of the BGC in five subfamilies. Phylogenetic analysis supports common ancestry of each enzyme class and enables reconstruction of a minimal ancestral cluster. We find that the BGC in *C. americana* has evolved bifunctionality, providing the scaffold of the formerly unidentified diterpene (+)-kaurene in addition to miltiradiene. This opens biosynthetic avenues towards previously inaccessible diterpenes in addition to highlighting an instance of BGC bifunctionality, which is rarely observed in plants[10,70]. We also discover complex miltiradiene BGCs in four additional species, laying the foundation for the elucidation of previously unknown diterpenoid pathways. Comparing the evolutionary trajectory of a BGC across a plant family illustrates how genomic organization can serve as a basis for expanding metabolic diversity.

## Results
### Genome assembly and annotation of *L. leonurus, P. barbatus,* and *P. vulgaris*

To increase the diversity of representatives across the Lamiaceae family, we sequenced three additional genomes, *Leonotis leonurus, Plectranthus barbatus*, and *Prunella vulgaris*, using the 10× Genomics linked read approach. High molecular weight DNA was isolated, 10× Genomics libraries constructed and Supernova was used to assemble the genomes generating pseudohaplotype assemblies; pseudohaplotype-1 was selected for downstream analyses resulting in 585 Mb (*L. leonurus*), 1.25 Gb (*P. barbatus*), and 820 Mb (*P. vulgaris*) assemblies (Table 1). For *P. barbatus* and *P. vulgaris*, the assembled genome size is consistent with the estimations of genome size from both flow cytometry, 1.53 Gb and 786 Mb, respectively, as well as from a k-mer-based estimation from Supernova, 1.29 Gb and 871 Mb, respectively (Supplementary Table 2). However, for *L. leonurus*, there was a discrepancy in genome size estimation between flow cytometry (1042 Mb), k-mers (688 Mb), and genome assembly (585 Mb). Coupled with the large distance between heterozygous SNPs in *L. leonurus* outputted from Supernova (16.9 Kb), it is most likely that *L. leonurus* is an autotetraploid and the Supernova assembly is representative of all homologous chromosomes.

Benchmarking Universal Single-Copy Ortholog (BUSCO)[71] of pseudohaplotype-1 assemblies revealed >97% complete BUSCOs in the three genomes (Table 2) with 18.5% and 13.4% duplicated BUSCOs present in *L. leonurus* and *P. barbatus*, respectively, suggesting of retained haplotigs in pseudohaplotype-1. Annotation of protein-coding genes with the unmasked genome using Lamiaceae-trained AUGUSTUS[72] matrices yielded 148,846 (*L. leonurus*), 413,222 (*P. barbatus*), and 229,613 (*P. vulgaris*) genes (Supplementary Table 3). Assessment of the completeness of the annotation using BUSCO with the predicted proteomes revealed 94.4% (*L. leonurus*), 92.2% (*P. barbatus*) and 91.2% (*P. vulgaris*) complete BUSCO orthologs, suggesting that the annotation provided a robust gene set. A total of 57.9%

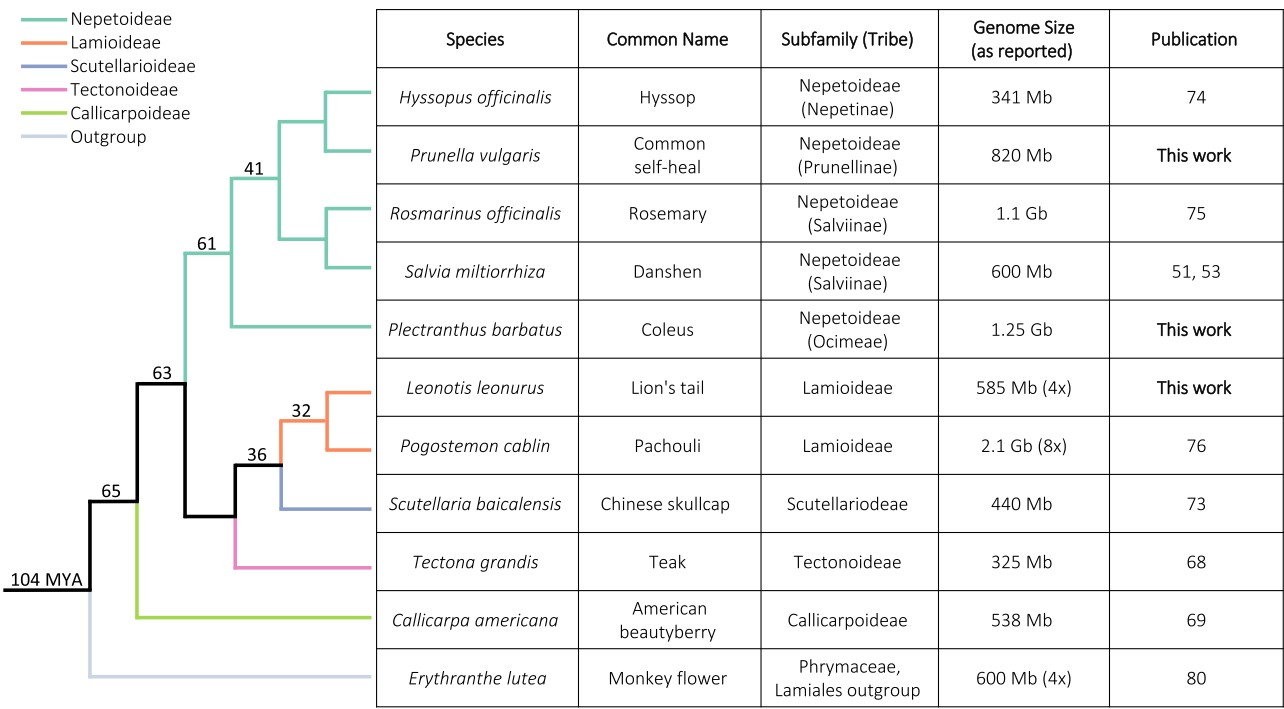

**Fig. 1 | Species and genome assemblies used in this study.** The cladogram shows evolutionary relationships between the species studied. Numbers at the nodes represent estimations of clade age in millions of years (MYA)[77–79]. Ploidy level of species not assumed to be diploid are shown in parenthesis next to their genome size (Supplementary Table 2).

**Table 1 | Statistics for the 10× Genomics assemblies of *Leonotis leonurus*, *Plectranthus barbatus*, and *Prunella vulgaris***

| | Number of scaffolds | Total size of scaffolds (bp) | N50 scaffold length (bp) | Number of Ns (Percent Ns) | Totals haps (consecutive Ns) | Largest scaffold (bp) |
|---|---|---|---|---|---|---|
| *Leonotis leonurus* | 23,651 | 585,264,293 | 1,094,942 | 40,883,810 (7.0%) | 15,483 | 11,593,990 |
| *Plectranthus barbatus* | 62,959 | 1,249,907,925 | 258,138 | 70,313,430 (5.6%) | 30,507 | 3,093,914 |
| *Prunella vulgaris* | 46,736 | 820,275,670 | 444,240 | 38,970,920 (4.8%) | 20,293 | 5,268,047 |

**Table 2 | Benchmarking Universal Single Copy Orthologs (BUSCOs) for *Leonotis leonurus*, *Plectranthus barbatus*, and *Prunella vulgaris* pseudohaplotype-1 genomes and predicted proteomes**

| | Species | Complete BUSCOs (C) | Complete single-copy BUSCOs (S) | Complete duplicate BUSCOs (D) | Fragmented BUSCOs (F) | Missing BUSCOs (M) |
|---|---|---|---|---|---|---|
| Genome | *Leonotis leonurus* | 98.5% | 80.0% | 18.5% | 0.5% | 1.0% |
| | *Plectranthus barbatus* | 97.8% | 84.4% | 13.4% | 1.0% | 1.2% |
| | *Prunella vulgaris* | 97.1% | 91.8% | 5.3% | 1.5% | 1.4% |
| Predicted proteome | *Leonotis leonurus* | 94.4% | 79.6% | 14.8% | 4.2% | 1.4% |
| | *Plectranthus barbatus* | 92.2% | 80.7% | 11.5% | 5.4% | 2.4% |
| | *Prunella vulgaris* | 91.2% | 86.8% | 4.4% | 6.1% | 2.7% |

(*L. leonurus*), 74.4% (*P. barbatus*), and 68.3% (*P. vulgaris*) of the genomes were repetitive with retroelements rather than DNA transposons dominating the genome space (Supplementary Table 4).

**Syntenic analysis reveals ubiquity of the miltiradiene biosynthetic gene cluster**

*C. americana* provided a unique opportunity to investigate the evolution of a family-wide diterpenoid BGC since it is in a sister lineage to the rest of the Lamiaceae and has a large, dense BGC. We analyzed nine

Lamiaceae genomes against our anchor species, *C. americana*, to determine synteny with its miltiradiene BGC. We chose our genome panel based on their assembly quality and contiguity as well as subfamily representation (i.e., phylogenetic placement). We chose three species with previously reported syntenic BGCs and available genomes (*S. miltiorrhiza*[24,51], *T. grandis*[68], and *S. baicalensis*[73]), the three species we assembled in this study (*L. leonurus*, *P. barbatus*, and *P. vulgaris)*, and three species with published genomes (*Hyssopus officinalis*[74], *R. officinalis*[75], and *Pogostemon cablin*[76]). In total, these ten species

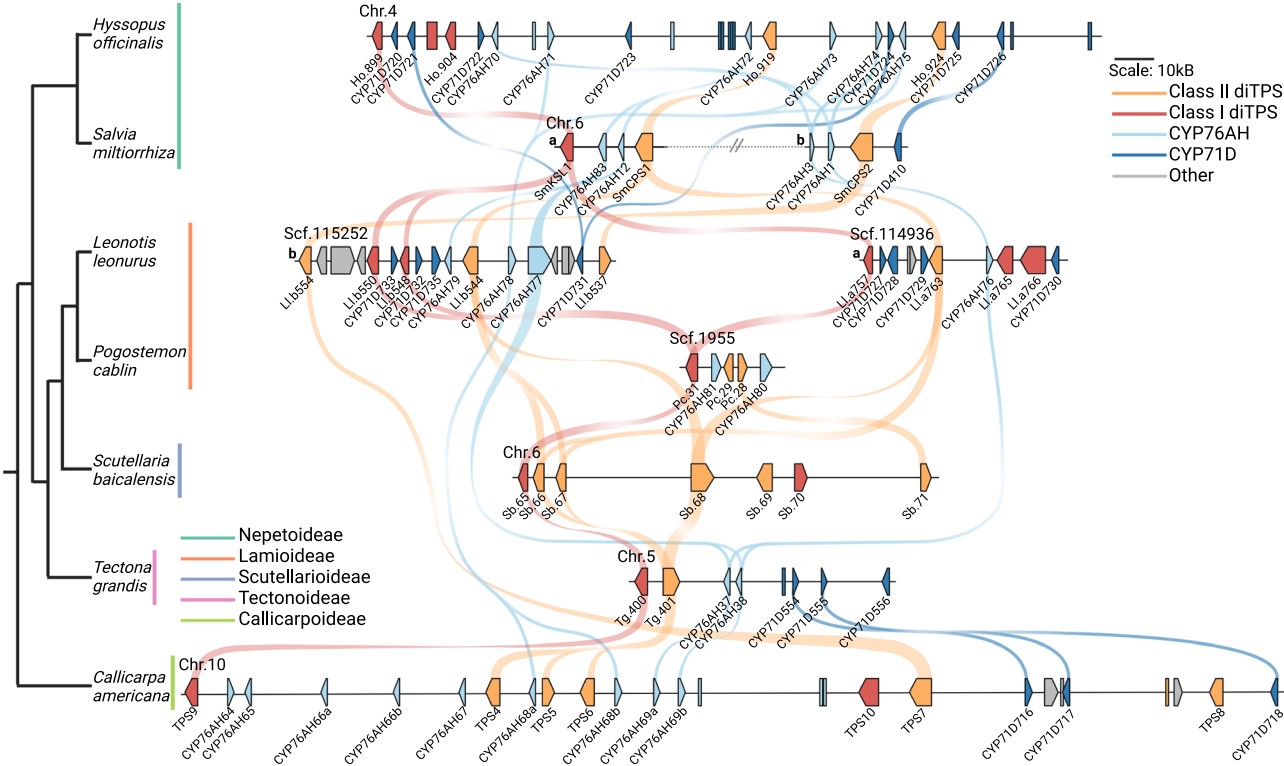

**Fig. 2 | Syntenic relationships of a miltiradiene biosynthetic gene cluster present across the Lamiaceae.** Genes are represented with arrows and pseudogenes are represented with boxes. A core set of genes are common to many species examined, including a diTPS class II (+)-CPP synthase, a diTPS class I miltiradiene synthase, and CYP450s in the 76AH and 71D subfamilies. Notably, there is divergence in gene number, cluster length, and unique genes, indicating lineage-specific evolution. Synteny between each species is shown here with colored curves. Species tree adapted from Mint Evolutionary Genomics Consortium 2018[124]. Figure created using BioRender.com. Source data are provided as a Source Data file.

represent five of the twelve currently recognized subfamilies with a most recent common ancestor estimated at 60–70 million years ago (Fig. 1)[77–79]. As a close Lamiales outgroup, we also analyzed *Erythranthe lutea* (Monkey flower; formerly *Mimulus luteus*)[80].

Out of the 10 Lamiaceae species sampled, all contained diTPS genes orthologous to known (+)-CPP and miltiradiene synthases. In seven species these diTPSs were within syntenic BGCs (Fig. 2). The genomes of *P. vulgaris*, *P. barbatus*, and *R. officinalis* were too fragmented to determine whether they were part of a larger cluster. To the best of our knowledge, four of the BGCs in this analysis have not been previously reported, showing that this cluster is even more conserved than originally described. All BGCs except that in *S. baicalensis* contain multiple CYP76AH genes. Five species, *C. americana*, *T. grandis*, *S. miltiorrhiza*, *H. officinalis*, and *L. leonurus*, also had at least one copy of a CYP71D gene.

Comparison of the BGCs provides insight into the formation and maintenance of this cluster in divergent lineages (Fig. 2). The *S. baicalensis* BGC uniquely contains no CYPs but appears to have tandem duplications of a class II diTPS and an additional non-syntenic class I diTPS. Non-syntenic diTPS and CYP genes are present in most of the BGCs, pointing toward dynamic assembly and independent refinement in each species. There are also several diTPS and CYP pseudogenes. Interestingly, there are few interrupting genes in these BGCs. The *H. officinalis* and *C. americana* BGCs encompass large genomic regions with more intergenic space, while others such as *P. cablin* and *L. leonurus* are compact and gene dense. We speculate that the presence of two clusters in *L. leonurus* is due to its tetraploidy and is not a true duplication. Similarly, octoploid *P. cablin* showed some evidence of multiple copies of the BGC (Supplementary Fig. 2). It is evident that each BGC, while maintaining the core miltiradiene genes, has undergone some lineage-specific independent evolution.

## Phylogenetic evidence of an ancestral miltiradiene cluster in Lamiaceae

To better understand the evolution of genes from each BGC, we estimated phylogenetic relationships for each enzyme subfamily in the BGCs along with a set of functionally characterized reference genes from Lamiaceae, except in the CYP71D clade where few characterized Lamiaceae sequences are available (Fig. 3). Consistent with other angiosperm labdane-type diTPSs, those diTPSs with class II function cluster in the TPS-c subfamily while those with class I function cluster in the TPS-e subfamily.

As expected, syntenic diTPSs in both subfamilies have common ancestry. Recent tandem duplications in the TPS-c family are evident in *C. americana* and *S. baicalensis* and contribute to lineage-specific BGC expansion (Figs. 3 and 4). The phylogenies also highlight the more distant origins of several non-syntenic diTPSs. The presence of divergent class I and II sequences points to independent acquisition as part of the diversification that occurred during speciation. Close inspection of phylogenetic relationships with characterized diTPSs can offer clues to likely functions. All class II diTPSs syntenic to *CamTPS6* phylogenetically cluster in clade TPS-c.2.2, which contains all known Lamiaceae (+)-CPP synthases as well as some diTPS enzymes which yield labdanes in the (+)-configuration. The two divergent class II enzyme sequences, Sb.71 and Pc.28, cluster in TPS-c.1 which produces compounds in the *ent*- rather than (+)-configuration, so it is likely that these two enzymes follow suit.

None of the class I enzymes encoded in the BGCs clustered in clade TPS-e.1, consistent with their expected role in specialized metabolism. The TPS-e.1 clade primarily contains enzymes that convert *ent*-CPP to the gibberellin intermediate *ent*-kaurene. All BGC class I diTPSs cluster in TPS-e.2, which mostly encodes enzymes that accept (+)-CPP as a substrate. The presence of a presumed (+)-CPP synthase

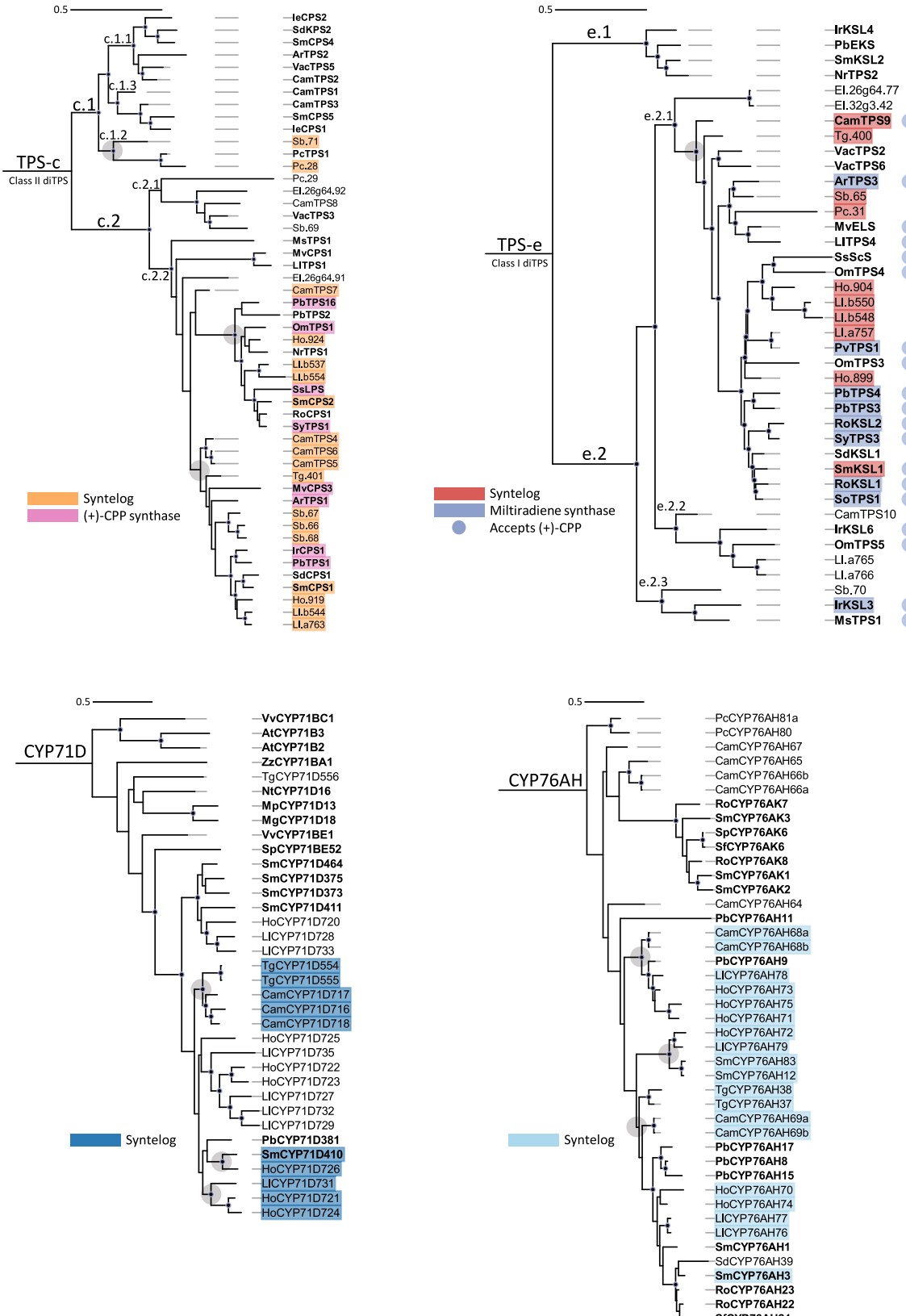

**Fig. 3 | Phylogenetic evidence shows the relatedness of each gene class in the clusters.** Enzymes present in each cluster with syntenic support from MCScanX and sequence identity from BLASTp are highlighted in red (TPS-e), orange (TPS-c), light blue (CYP76AH), and dark blue (CYP71D). DiTPSs characterized in previous reports are highlighted in pink and periwinkle ((+)-CPP synthases for TPS-c and miltiradiene synthases for TPS-e, respectively). Reference enzymes are bolded.

Black solid dots at the base of the nodes represent 80% bootstrap confidence. Gray circles around clade nodes represent hypothetical expansion points for syntelogs and share approximately 70% or more sequence similarity. DiTPS trees are rooted to *Physcometrium patens ent*-kaurene synthase (PpEKS), and CYP trees are rooted to *Arabidopsis thaliana* AtCYP701A3. Source data are provided as a Source Data file.

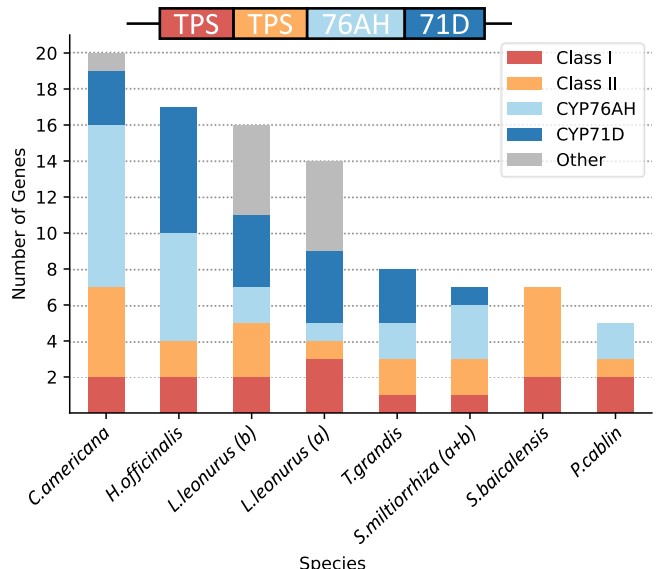

**Fig. 4 | Predicted Lamiaceae minimal ancestral BGC and species-specific expansion of each.** Based on maximum parsimony, we suggest that a cluster containing a class II diTPS, class I diTPS, CYP76AH, and CYP71D gene was formed in an early Lamiaceae ancestor. Lineage-specific expansion and refinement are evident from the number and composition of genes in each gene family present in the extant species studied. Source data are provided as a Source Data file.

encoded in every BGC supports the likelihood that these class I diTPSs can all utilize (+)-CPP. Genes syntenic with *CamTPS9* are grouped in clade TPS-e.2.1, which contains all but one of the Lamiaceae sequences encoding enzymes known to catalyze formation of miltiradiene. Notably, every BGC contains at least one of these presumed miltiradiene synthase sequences. Also characteristic of the TPS-e.2.1 clade is the loss of the internal γ domain, which is retained in most diTPSs but lost in mono- and sesqui-TPSs. The three non-syntenic enzyme sequences are split between clades TPS-e.2.2 and TPS-e.2.3, which encompass only a few characterized sequences with unique functions. The functional heterogeneity of these clades makes it difficult to draw conclusions as to the likely function of these BGC encoded enzymes but does offer intriguing possibilities for discovery of novel terpene backbones.

While phylogenetic classification is not a perfect predictor of TPS function[37,81], previous work has demonstrated a high level of clade specific consistency that allows us to draw tentative conclusions about the function of the BGC diTPSs[48]. Phylogenetic evidence supports that these BGCs likely have at minimum a (+)-CPP synthase and a miltiradiene synthase, enabling production of miltiradiene in each plant (Fig. 3). Moreover, several BGCs contain diTPSs from clades that may offer distinctive chemistries.

CYPs in the 76AH enzyme subfamily exhibit close phylogenetic clustering across the species analyzed. Several functionally characterized CYP76AHs have been found to oxidize miltiradiene in critical steps towards tanshinone and carnosic acid biosynthesis[54,55]. Although we were unable to identify a BGC in *R. officinalis* due to a fragmented assembly, the close relationship between the RoCYP76AH enzymes and those the other BGCs supports common ancestry. Nearly all CYP76AHs in the BGCs have paralogs within each cluster, highlighting the role of tandem duplication in expanding this subfamily[46,82]. However, there are several BGC CYP76AHs that are highly divergent from the syntelogs. The *C. americana* enzymes CYP76AH65, CYP76AH66, and CYP76AH67 are phylogenetically distinct, showing only 50–60% sequence similarity to other BGC CYP76AHs. These enzymes are more related to the clade of CYP76AKs, which have not been found in this

BGC but are part of the tanshinone and carnosic acid oxidation networks.

CYPs in the 71D subfamily similarly show phylogenetic clustering with others in the BGCs. Three CYP71D sequences from *H. officinalis* and *L. leonurus* are in the same clade as the CYP71D gene array from *S. miltiorrhiza*, which was implicated in furan ring formation for the tanshinones[24]. SmCYP71D410 is a member of the BGC Sm-b that phylogenetically clusters with HoCYP71D724 and PbCYP71D381 enzymes. PbCYP71D381 can oxidize the forskolin precursor (13R) manoyl oxide, a close structural relative of miltiradiene[83]. One enzyme from *T. grandis* stands out as much less related than the rest, with only 40–50% sequence similarity to other BGC CYP71Ds. This enzyme is likely another recent independent acquisition, although it is the only one observed in the CYP71D subfamily. All BGCs containing CYP71D genes also have at least one duplication, once again highlighting the importance of duplication in the diversification of these pathways[84].

Close phylogenetic clustering of most enzymes in all four subfamilies provides compelling evidence for a common ancestral origin and subsequent lineage-specific duplications. We analyzed presence/absence of syntelogs and proposed a model for a minimal cluster using ancestral state reconstruction (Fig. 4; Supplementary Figs. 3, 4). High levels of sequence conservation between syntelogs supports a minimal ancestral cluster that contains genes encoding a (+)-CPP synthase, a miltiradiene synthase, a CYP76AH, and a CYP71D. The dynamic nature of this BGC over millions of years of evolution is evident through the gene loss, presence of pseudogenes, and addition of non-syntenic genes observed in these extant Lamiaceae. Despite these differences, the high degree of conservation of the ancestral cluster is notable.

Since the miltiradiene BGC was present in nearly every Lamiaceae species sampled, we also investigated the synteny in *E. lutea*, a closely related Lamiales outgroup[77,80,85]. We found a region syntenic to the *C. americana* BGC which contains class II and class I diTPSs but no CYPs (Supplementary Fig. 5). The genes encoding class II enzymes, *El.26g64.91* and *El.26g64.92*, are in clade TPS-c.2, showing some similarity with other (+)-CPP synthases (Fig. 3). The class I sequence, *El.26g64.77*, is within TPS-e.2.1, but distinct from the rest of the clade, and surprisingly retains the γ domain. This domain loss has occurred multiple times in the evolution of plant TPSs[86], so it is conceivable that the class I enzymes in *E. lutea* represent the three-domain miltiradiene synthase shared by the most recent common ancestor in the Lamiales. While the *E. lutea* partial cluster may provide a glimpse into an ancestral state of the Lamiaceae BGC, a more widespread examination of additional Lamiales genomes would be an interesting avenue for future work and could more firmly establish the timeline of gene acquisition and loss in this cluster.

## Functional characterization of the *C. americana* BGC reveals two metabolic modules and a terpene backbone

Though increasing numbers of computationally predicted BGCs have been identified in plants, only a few are functionally characterized. So far, coregulation has proven to be a greater predictor of functional relationship in BGCs than colocalization alone[87]. Previous analysis of the two BGCs in *S. miltiorrhiza*, Sm-a and Sm-b, found that each had divided expression between root and aerial tissues. The diTPSs from Sm-a and *CYP76AHs* from Sm-b were expressed exclusively in root tissues and found to be vital steps in the root tanshinone biosynthetic pathway[50]. Additionally, an array of root-specific CYP71D encoded enzymes were also integral to tanshinone biosynthesis but located elsewhere in the genome[24]. Another example where differentially expressed diTPSs and CYPs were reported in distinct specialized metabolite pathways despite being colocalized is the bifunctional gene clusters of phytocassanes/oryzalides found in *Oryza sativa* (rice)[70] and the noscapine/morphinan biosynthesis in *Papaver ssp.* (poppy)[11,57]. Divergence in expression may be one way in which plants exploit some

of the benefits of genomic organization while creating unique pathways based on regulation.

Given the unprecedented size and complexity of the BGC identified in *C. americana*, we sought to investigate whether it is a metabolically unified BGC. We first analyzed RNA expression in 8 tissue types to determine the expression pattern of the BGC (Fig. 5; Supplementary Fig. 6)[69]. This revealed a clear divergence between the first and second halves of this BGC. The first half is preferentially expressed in fruit and root tissue and contains a (+)-CPP synthase (*CamTPS6*)[69], the predicted miltiradiene synthase (*CamTPS9*), and several CYP76AHs. The second half is more strongly expressed in flower and young leaf tissues and contains a non-orthologous class I diTPS (*CamTPS10*), another predicted (+)-CPP synthase (*CamTPS7*), and two CYP71Ds as well as partial fragments of a CYP76AH (*Ca.26-27*). The presence of a diTPS class II/ class I pair as well as CYPs in each module suggests that this BGC may have evolved divergent diterpenoid pathways. Additionally, we investigated expression of each of BGC in the other species with published transcriptomic data but found no overarching expression trends, unlike in *C. americana* (Supplementary Fig. 6).

We investigated enzyme activity for the following members of the *C. americana* cluster: *CamTPS7, CamTPS8, CamTPS9, CamTPS10, CamCYP76AH64, CamCYP76AH6S, CamCYP76AH67, CamCYP76AH68, CamCYP76AH69, CamCYP71D716*, and *CamCYP71D717*. Combinations of all genes were transiently expressed in *Nicotiana benthamiana* to evaluate enzyme function. DiTPS functions were determined by comparison of mass spectra and retention time by GC-MS with published diTPS activities or using NMR for previously unpublished activity (Fig. 6). CamTPS7 was confirmed to be a (+)-CPP synthase (Supplementary Fig. 7). CamTPS9 is a miltiradiene (**1**) synthase, with some abietatriene (**2**; *ent*-abieta-8,11,13-triene) resulting from spontaneous aromatization *in plantae* consistent with previous observations[88]. CamTPS10, when paired with a (+)-CPP synthase, forms (+)-kaurene (**4**) (Supplementary Figs. 8–10). The biological relevance of this activity is supported by the structure of the diterpenoid calliterpenone, which is derived from the (+)-kaurene backbone and has been documented in multiple *Callicarpa* species[89]. Calliterpenone has been investigated for its potential as a plant growth-promoting agent[90], and thus represents an interesting biosynthetic target. Discovery of this (+)-kaurene synthase will enable biosynthetic access to this group of metabolites as well as to non-natural diterpenoids that may have useful bioactivities[91]. The physical grouping and similar expression patterns of *CamTPS10* and *CamTPS7* supports that this cluster has diverged into two metabolically distinct modules through the duplication of a (+)-CPP

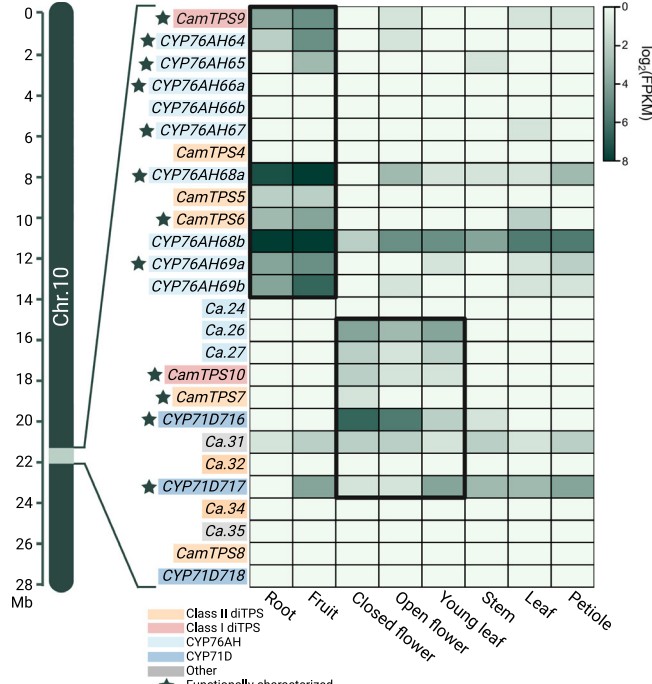

**Fig. 5 | Tissue specific expression of a miltiradiene BGC in *C. americana* obtained from RNA sequencing.** Functional characterization of these enzymes refers to this study. This figure represents Chr10:21.92-22.33 Mb. Approximate location on the chromosome is indicated. Two differentially expressed metabolic clusters are boxed to highlight similar expression patterns. Colors indicate diTPS, CYP, or unrelated gene family, including pseudogenes (unnamed). Data obtained from Hamilton et al.[69].

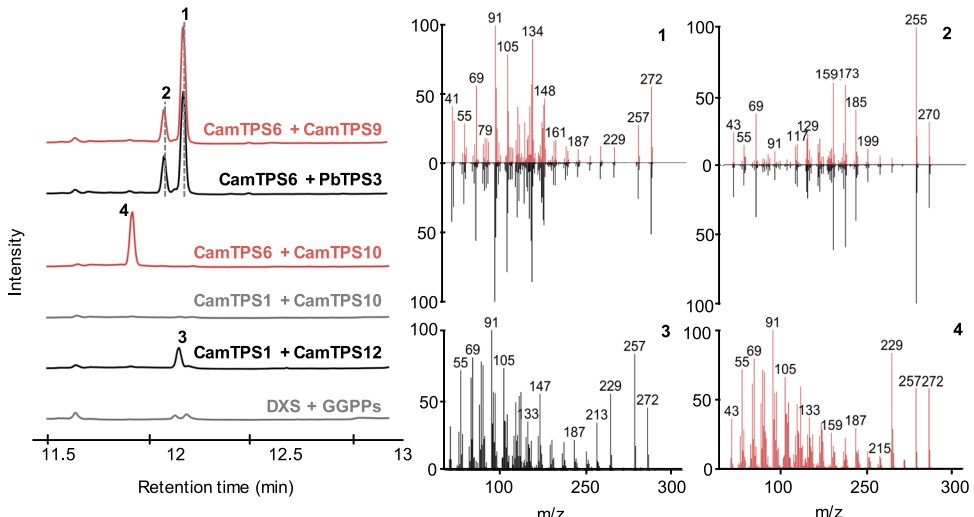

**Fig. 6 | GC-MS analysis of *C. americana* BGC diTPS products.** CamTPS9 was confirmed to be a miltiradiene synthase by comparison with the retention time and mass spectra of PbTPS3[125–128] products when both were expressed with the (+)-CPP synthase CamTPS6[69], forming miltiradiene (**1**) and abietatriene (**2**). CamTPS10 was found to make **4** from (+)-CPP but not *ent*-CPP (CamTPS1)[69]. This product has a different retention time but similar mass spectrum to *ent*-kaurene (**3**), made by the combination of CamTPS1 and CamTPS12 (Supplementary Fig. 11). All chromatograms shown are total ion chromatograms. Red and black traces correspond to combinations yielding **1**, **2**, **3**, and **4** respectively, as indicated in the mass spectra. Each combination includes *P. barbatus* 1-deoxy-D-xylulose-5-phosphate synthase (*DXS*) and GGPP synthase (*GGPPS*), shown as a control in gray. Source data are provided as a Source Data file.

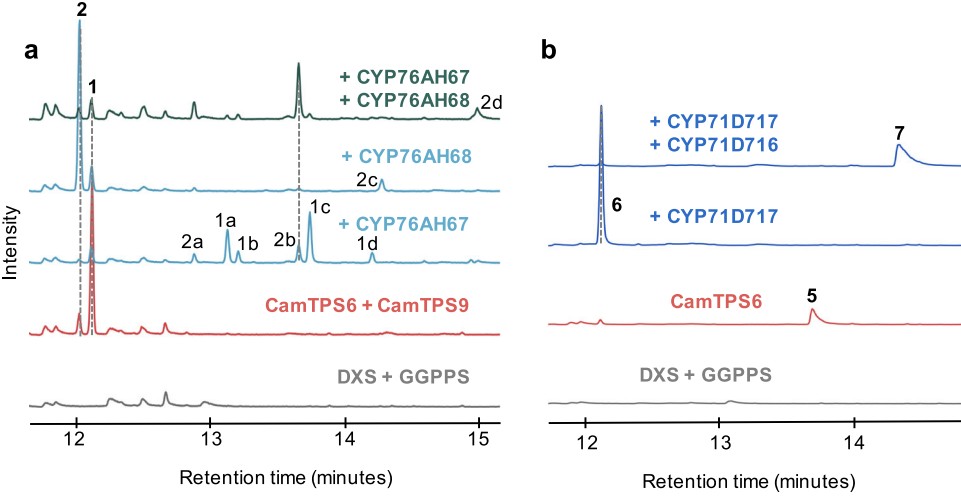

**Fig. 7 | GC-MS chromatograms showing oxidation products of *C. americana* BGC CYPs. a** Oxidation products of the CamCYP76AHs from **1** and **2**, assigned based on analysis of mass spectra (Supplementary Fig. 12). **b** CamCYP71D717 catalyzes the production of (+)-manool (**6**), likely from (+)-copalol (**5**) (Supplementary Fig. 16,) and the addition of CamCYP71D716 results in 3(*S*)-hydroxy-(+)-manool (**7**).

Each combination includes *P. barbatus* 1-deoxy-D-xylulose-5-phosphate synthase (*DXS*) and GGPP synthase (*GGPPS*), shown as a control in gray. CamTPS6 and CamTPS6 + CamTPS9 controls given in red. Source data are provided as a Source Data file.

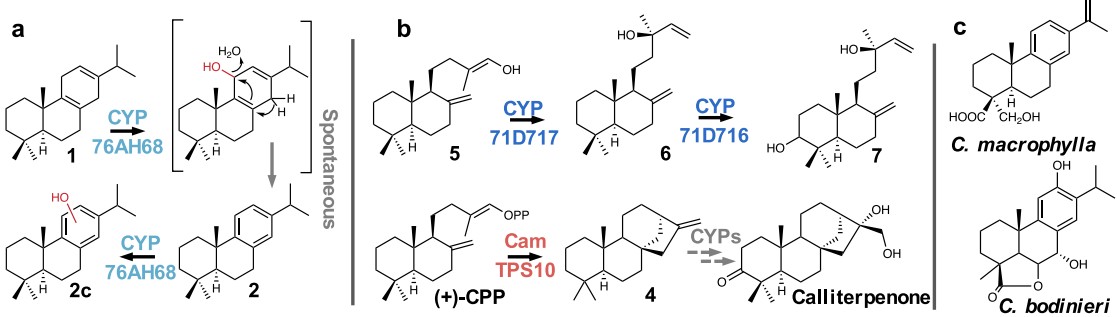

**Fig. 8 | Pathway schematic for CYP oxidations in *C. americana*. a** Proposed mechanism for enzyme-assisted conversion of **1** to **2**, followed by an additional oxidation of **2** to form **2c**. Mass spectra supports assignment of the hydroxy group in **2c** to the c-ring (Supplementary Fig. 12). **b** Proposed conversion of **5** to **6** by

CamCYP71D717, and oxidation of **6** by CamCYP71D716. This occurs in the same position as a keto group on calliterpenone, which is derived from **4**. **c** Structures of abietane diterpenoids found in two other species of *Callicarpa*.

synthase, the recruitment of an additional class I diTPS, and a shift in tissue-specific gene expression.

After establishing routes to the formation of the *C. americana* diterpene backbones, we tested each CYP against all possible diterpene intermediates found in this plant (Fig. 7): *ent*-kaurene (CamTPS12; Supplementary Fig. 11) and kolavenol[69] formed by diTPSs outside the cluster, and (+)-kaurene and miltiradiene from the BGC. No activity was detected with kolavenol or *ent*-kaurene. With miltiradiene, CamCYP76AH67 formed six different oxidation products (**1a–d**, **2a–b**, Fig. 7a). Based on m/z of the molecular ions and comparison of mass spectra with each other and the NIST database, two match oxidations of abietatriene and the other four of miltiradiene (Supplementary Fig. 12). Most of these products proved difficult to separate by column chromatography, preventing complete structural elucidation. However, we were able to purify **2a**, and NMR experiments support the assignment as 15-hydroxy-*ent*-abieta-8,11,13-triene (Supplementary Figs. 13–15). Oxidation in this position on an abietane diterpene has only been reported twice before: by a 2-oxoglutarate dehydrogenase in *S. miltiorrhiza*[92] and by CYP81AM1 in *Tripterygium wilfordii*[93]. CamCYP76AH68 also showed activity with miltiradiene, dramatically shifting the product profile towards abietatriene and affording a small amount of oxidized abietatriene (**2c**; Supplementary Fig. 12). This indicates that CamCYP76AH68 may be hydroxylating the c-ring of miltiradiene, which then undergoes water

loss to form abietatriene more readily than the spontaneous aromatization of miltiradiene alone (Fig. 8a). In previous work characterizing enzymes involved in tanshinone and carnosic acid biosynthesis, the ferruginol synthases showed a preference for abietatriene, but enzymatic conversion of miltiradiene to abietatriene was not observed. It was suggested that the aromatization is spontaneous and possibly driven by sunlight[88]. The discovery of CamCYP76AH68 indicates that at least in *C. americana* an enzyme may assist in the conversion of miltiradiene to abietatriene. When we expressed each CYP with *CamTPS6* and *CamTPS10* to evaluate CYP activity with the (+)-kaurene backbone, we observed a peak with expression of *CamCYP71D717*. Upon further investigation, however, we realized this enzyme apparently catalyzes formation of (+)-manool (**6**) from (+)-copalol (**5**), the dephosphorylation product of (+)-CPP (Fig. 7b and Supplementary Fig. 16). Each CYP/TPS enzyme combination that resulted in observable products was then expressed in combination with all other CYPs. *CamCYP76AH67* combined with *CamCYP76AH68* and miltiradiene yielded at least one oxidized compound (**2d**, Fig. 7a; Supplementary Fig. 12). The combination of *CamTPS6* with *CamCYP71D716* and *CamCYP71D717* resulted in full conversion of (+)-manool (**6**) to 3(*S*)-hydroxy-(+)-manool (**7**), which was confirmed by NMR (Figs. 7b, 8b; Supplementary Figs. 17–19).

To the best of our knowledge, no abietane-type diterpenoids were previously reported in *C. americana*, which has been primarily studied

for clerodane diterpenoids produced in leaves[94–96]. However, other *Callicarpa* species, including *C. bodinieri* and *C. macrophylla*[97], produce a wide variety of medicinally relevant abietane diterpenoids (Fig. 8c), indicating that the abietane skeleton is a key intermediate for at least some plants in this genus[64,97]. We analyzed a whole root extract of *C. americana* by GC-MS and found compounds with matching retention time and mass spectra to abietatriene and the oxidized product (**2c**) produced by CYP76AH68. This supports the biological relevance of enzyme activities elucidated in *N. benthamiana* (Supplementary Fig. 20).

*C. americana* contains over 600 predicted CYPs, and it is likely that the BGC CYPs are part of a larger metabolic network with peripheral modifying enzymes elsewhere within the genome[69]. However, the functional activities we report here validate the biological significance of the BGC and its divergent modules. The CYPs showed a marked preference for the (+)-copalol and miltiradiene backbones over other diterpenes present in the plant. Within the two modules, the miltiradiene and (+)-kaurene synthases were differentially expressed along with their respective (+)-CPP synthases. The CYP76AHs were more active towards miltiradiene, whereas the CYP71Ds utilized (+)-copalol. Functionalization of (+)-kaurene may require oxidations catalyzed by non-clustered enzymes.

## Discussion

In this study we found that the miltiradiene BGC, previously identified in only a few species, is present across five divergent Lamiaceae subfamilies. The preserved enzyme sequences and gene order in the cluster provide strong evidence for an ancestral cluster in an early Lamiaceae ancestor. From this core cluster, these species have retained the diTPSs necessary to form the signature miltiradiene backbone but tailored their chemical diversity through gene duplication, sequence divergence, gene acquisition, and gene loss. We can speculate that the metabolic products from the ancestral cluster have diversified as the Lamiaceae family diverged and populations adapted to new environments. Gene duplication appears to be a major driver of the evolution and expansion of the vast diversity of TPSs and CYPs in plants[2,41,86,98], and the Lamiaceae miltiradiene cluster exemplifies this. This is notable in the *C. americana* cluster where tandem duplication has generated five sequential, highly similar CYP76AH genes. However, every species examined had at least one apparent duplication event, supplying the material for evolution toward metabolic diversification. There is also a striking example of cluster expansion through the apparent recruitment of *CamTPS10* in *C. americana*. The discovery of the (+)-kaurene synthase showcases another example of a bifunctional BGC with divergent transcription patterns. The presence of phylogenetically distinct diTPSs in other here discovered miltiradiene BGCs similarly suggests multifunctionality.

Conservation of the miltiradiene backbone suggests strong selective pressures for retention in the Lamiaceae and beyond, as illustrated by the clustered pair of diTPSs and orthologs of recently discovered CYPs forming and oxidizing the same backbone in *T. wilfordii* in the distant Celastraceae[99]. Surprisingly little is known about how plants use abietane diterpenoids, but they are mostly thought to be involved in pathogen responses due to their antibacterial activities[60,100]. However, abietanes have been extensively studied for their importance to human health. They exhibit a range of bioactivities from anti-tumor to antimicrobial to anti-inflammatory, among others[60–63,101]. Nearly 500 abietane diterpenoids have been reported to date in Lamiaceae species[40,102]. Earlier investigations of these diterpenoids in Lamiaceae have taken a metabolite-guided approach, which has yielded much progress towards the biosynthesis of tanshinones, carnosic acid, and related compounds. The findings of this study establish a framework for a genomics-guided investigation of additional abietane diterpenoids throughout the Lamiaceae. The functional characterization of part of the *C. americana* BGC as well as the root metabolite data support the presence of a miltiradiene diterpenoid network in this plant despite the lack of previously documented abietanes. Further characterization of the other identified miltiradiene BGCs in *H. officinalis*, *P. cablin*, and *L. leonurus* could similarly lead to the discovery of yet unknown chemistries.

A deeper understanding of the enzymatic activities encompassed by BGC genes will also help to elucidate how BGCs drive expansion of metabolic diversity. It is clear from the conservation of the miltiradiene BGC in at least five extant Lamiaceae subfamilies that gene colocalization is an important contributor to plant specialized metabolism. Genomic organization is also of special interest in synthetic biology, as understanding natural BGCs can provide a blueprint for the construction and control of synthetic clusters in heterologous systems[103]. This study presents one of the currently limited examples of a BGC present throughout an entire family. With the increasing quality and quantity of plant genomes available, future large-scale BGC investigations may find that plants frequently rely on BGCs as a toolbox for adaptability through metabolic diversity.

## Methods

### Collinearity analysis

The BLAST function makeblastdb (*E*-value of 1e⁻¹⁰, 5 alignments)[104] was used to create protein databases between *C. americana* and each other species examined. Peptide sequences and genome annotation files were obtained through respective data repositories. Syntenic analysis between *C. americana* and every other species discussed was performed using the standard MCScanX pipeline (Match score = 50; Match size = 5; Gap penalty = −1; Overlap window = 5; *E*-value = 1e⁻⁵; Max gaps = 25)[105]. Results were visualized using SynVisio[106]. Orthologs and syntenic lines were manually curated using 70% sequence identity cutoff determined by the BLASTp alignment function (Threshold = 0.05, Word Size = 3, Matrix = BLOSUM62, Gap Costs = Existence:11 Extension:1).

### Ancestral state reconstruction

Extant character states were collected into a single document coded as 1 for presence and 0 for absence of each gene. Ancestral state analysis was performed using the phytools R package (version 0.7-80)[107]. Evolutionary models were selected using information from the 'fitMK()' function. Ancestral states were determined with the 'ace()' function.

### Phylogenetic trees

Sequences used in all protein phylogenies were obtained from annotated peptide sequences from their respective species. A list of reference sequences used can be found in the source data of Fig. 2. CYP annotation was kindly provided by David Nelson (University of Tennessee). Full-length protein coding sequences were used, however plastidial targeting sequences present in diTPSs were removed from alignments. Multiple sequence alignments were generated using ClustalOmega (version 1.2.4; default parameters)[108] and phylogenetic trees were generated by RAxML (version 8.2.12; Model = protgammaauto; Algorithm = a)[109] with support from 1000 bootstrap replicates. All alignments are available in our dryad repository (https://doi.org/10.5061/dryad.w9ghx3frg).The tree graphic was rendered using the Interactive Tree of Life (version 6.5.2)[110].

### Genome sequencing, assembly, and annotation of three Lamiaceae species

High molecular weight DNA was isolated from mature leaves from *L. leonurus*, *P. barbatus*, and *P. vulgaris* and used to construct a 10× Genomics library using the Genome and Gel Bead Kit v2 (10× Genomics, Pleasanton, CA). Libraries were sequenced on an Illumina NovaSeq 6000 (Illumina, San Diego, CA) in paired end mode, 150 nt. Libraries were made and sequenced by the Roy J. Carver Biotechnology Center at the University of Illinois at Urbana-Champaign. The genomes

were assembled using 10× Supernova (version 2.1.1)[111]. The script 'supernova run' was run with default settings except–maxreads was set to 360000000 (*P. vulgaris*), 531000000 (*P. barbatus*) or 297550000 (*L. leonurus*), which yielded the best results for genome contiguity and percent of estimated genome size after testing multiple coverage levels. To obtain fasta files, 'supernova mkoutput' was run with the parameters, '–style=pseudohap2' and '–headers=full'. Genes were predicted on the non-repeat-masked pseudohaplotype-1 assemblies using AUGUSTUS (version 3.3)[72] with the parameter, '–UTR = off', and the '–species' and 'c–extrinsicCfgFile' parameters to use training results from closely related species, *H. officinalis* (*P. barbatus*, *P. vulgaris*) or *T. grandis* (*L. leonurus*). Assembly statistics were calculated using the tool assembly-stats (version 1.0.1)[112]. The AUGUSTUS default gene annotations were converted to GFF3 format using the gtf2gff.pl in the AUGUSTUS repository (version 3.4.0) and gene annotation metrics were generated using GAG (version 2.0.1)[113]. BUSCO (version 5.2.2)[71] was run in genome mode using the lineage dataset 'embryophyta_odb10.' To identify repetitive sequences in the three de novo assembled genomes, a custom repeat library (CRL) for each assembly was created with RepeatModeler (version 2.0.3)[114]. Protein-coding genes were removed from each CRL using ProtExcluder (version 1.2)[115] and RepBase Viridiplantae repeats from RepBase (version 20150807)[116] were added to create a final CRL. Each assembly was repeat masked with its corresponding CRL using RepeatMasker (version 4.1.2-p1)[117] using the parameters -e ncbi -s -nolow -no_is -gff.

### Transcriptomic analysis
All transcriptomic datasets used in Fig. 5 and Supplementary Fig. 6 were downloaded from the SRA database. Raw reads were trimmed using fastp (version 0.23.2)[118], mapped to respective coding sequence files using Salmon 'index' (version 1.8.0)[119], and quantified using Salmon 'quant' (libtype=A, validate mappings). Genes specific to each respective cluster were parsed out to compare expression levels between tissues. Data was transformed by a factor of log2(X + 1), where the quantified expression, X, had a value of 1 added to all genes in an unbiased fashion to account for occurrences of 0 expression and to remove negative log values due to lowly expressed genes, which would exaggerate differences between genes. The caveat to this transformation is lower expressed genes appear to have expression closer to 0 while more highly expressed genes are comparatively unaffected. Genes were clustered based on order of appearance within the genome, while tissues were clustered based on the similarity between tissue groups. Heatmaps were generated using ggplot2 (version 3.1.1)[120].

### PCR and cloning
Synthetic oligonucleotides are given in Supplementary Table 5, GenBank accession numbers, and sequences of all enzymes characterized or discussed in this study are listed in the source data of Figs. 2 and 3. Candidate enzymes were PCR-amplified from root, fruit, leaf, and flower cDNA, and coding sequences were cloned and sequence-verified with respective gene models. Constructs were then cloned into the plant expression vector pEAQ-HT[121] and used in transient expression assays in *N. benthamiana*.

### Transient expression in *N. benthamiana*
Transient expression assays in *N. benthamiana* were carried out based on a published protocol[48]. Specifically, *N. benthamiana* plants were grown for 5 weeks in a controlled growth room under 16 H light (24 °C) and 8 H dark (17 °C) cycle before infiltration. Constructs for co-expression were separately transformed into *Agrobacterium tumefaciens* strain LBA4404. Cultures were grown overnight at 30 °C in LB with 50 μg/mL kanamycin and 50 μg/mL rifampicin. Cultures were collected by centrifugation and washed twice with 10 mL water. Cells were resuspended and diluted to an $OD_{600}$ of 1.0 in water with 200 μM

acetosyringone and incubated at 30 °C for 1–2 H. Separate cultures were mixed in a 1:1 ratio for each combination of enzymes, and 4–5 week old plants were infiltrated with a 1 mL syringe into the underside (abaxial side) of *N. benthamiana* leaves. All gene constructs were co-infiltrated with two genes encoding rate-limiting steps in the upstream 2-C-methyl-D-erythritol 4-phosphate (MEP) pathway: *P. barbatus* 1-deoxy-D-xylulose-5-phosphate synthase (*PbDXS*) and GGPP synthase (*PbGGPPS*) to boost production of the diterpene precursor GGPP[91,122]. Plants were returned to the controlled growth room (24 °C, 12 H diurnal cycle) for 5 days. Approximately 200 mg fresh weight from infiltrated leaves was extracted with 1 mL hexane (diTPS products) or ethyl acetate (CYP products) overnight at 18 °C. Plant material was collected by centrifugation, and the organic phase was removed for GC-MS analysis. Each experiment was performed in triplicate. Data shown are from single experiments representative of the replicates.

### Root metabolite extraction
The entire root system of a healthy 3 year old C. americana plant grown under greenhouse conditions was collected, washed, and blended with water to break down the tissue. The mixture was then combined with 500 mL ethyl acetate and allowed to extract for 24 H. The organic layer was then separated from the aqueous layer, filtered, concentrated via rotary evaporator, and stored at −20 °C. This extract was diluted 1:500 in ethyl acetate and analyzed by GC-MS. All GC-MS analyses were performed in Michigan State University's Mass Spectrometry and Metabolomics Core Facility on an Agilent 7890 A GC with an Agilent VF-5ms column (30 m × 250 μm × 0.25 μm, with 10 m EZ-Guard) and an Agilent 5975 C detector. The inlet was set to 250 °C splitless injection of 1 μL and He carrier gas (1 mL/min), and the detector was activated following a 3 min solvent delay. All assays and tissue analysis used the following method: temperature ramp start 40 °C, hold 1 min, 40 °C/min to 200 °C, hold 4.5 min, 20 °C/min to 240 °C, 10 °C/min to 280 °C, 40 °C/min to 320 °C, and hold 5 min. MS scan range was set to 40–400.

### Product scale-up and NMR
For NMR analysis, production in the *N. benthamiana* system was scaled up to 1 L. A vacuum-infiltration system was used to infiltrate *A. tumefaciens* strains in bulk. *N. benthamiana* leaves. Approximately 80 g of leaf tissue was extracted overnight in 600 mL hexane at 4 °C and 150 rpm. The extract was dried down on a rotary evaporator. Each product was purified by silica gel flash column chromatography with a mobile phase of 100% hexane for (+)-kaurene and successive column washes from 100% hexane to 95/5 hexane/ethyl acetate for 3(*S*)-hydroxy-(+)-manool. NMR spectra were measured in Michigan State University's Max T. Rogers NMR Facility on a Bruker 800 MHz or 600 MHz spectrometer equipped with a TCl cryoprobe using $CDCl_3$ as the solvent. $CDCl_3$ peaks were referenced to 7.26 and 77.00 ppm for $^1H$ and $^{13}C$ spectra, respectively.

### Reporting summary
Further information on research design is available in the Nature Portfolio Reporting Summary linked to this article.

## Data availability
The data supporting the findings of this work are available within the paper and the Supplementary Information files. The raw genomic reads generated in this study have been deposited in the NCBI BioSample database under the following accession codes: *Plectranthus barbatus* (SAMN26547115), *Leonotis leonurus* (SAMN26547116), and *Prunella vulgaris* (SAMN26547117). The genome assemblies have been deposited in NCBI with the following accession codes: *Plectranthus barbatus* (JAPKLW000000000 [https://www.ncbi.nlm.nih.gov/nuccore/JAPKLW000000000.1/]), *Leonotis leonurus* (JAPKLX000000000 [https://www.ncbi.nlm.nih.gov/nuccore/JAPKLX000000000.1/]), and *Prunella vulgaris* (JAPKLY000000000 [https://www.ncbi.nlm.nih.gov/nuccore/JAPKLY00

0000000.1/]). The versions described in this paper are versions XXXXXX010000000. Sequences for the functionally characterized enzymes from *Callicarpa americana* can be found in the NCBI GenBank database: ON260868 [https://www.ncbi.nlm.nih.gov/nuccore/ON260868.1/]-ON260876 [https://www.ncbi.nlm.nih.gov/nuccore/ON260876.1/]. Additional data including genome assemblies and annotations, GC-MS raw data, NMR raw data, phylogenetic alignments, cluster sequences, and collinearity files can be found in our Dryad Repository [https://doi.org/10.5061/dryad.w9ghx3frg].[123] Source data are provided with this paper.

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

## Acknowledgements

This work was supported in part through computational resources and services provided by the Institute for Cyber-Enabled Research at Michigan State University and the Georgia Advanced Computing Resource Center. We would like to thank Dr. Cassandra Johnny of Michigan State University's Mass Spectrometry and Metabolomics Core Facility for their help in obtaining and interpreting GC-MS data, and Dr. Daniel Holmes and the Max T. Rogers NMR Facility for their help in obtaining and interpreting NMR data. We would also like to thank Dr. David Nelson for the naming of all CYP sequences presented in this work, Dr. Kevin Childs for his advice and guidance, Dr. Wajid Bhat for extracting DNA for genomic sequencing, and Malik Sankofa for assistance with plants, media, and general lab preparation. This work was supported by the Michigan State University Strategic Partnership Grant program ('Evolutionary-Driven Genome Mining of Plant Biosynthetic Pathways') to B.H. and C.R.B. and through Georgia Research Alliance funds to C.R.B. B.H. gratefully acknowledges the US Department of Energy Great Lakes Bioenergy Research Center Cooperative Agreement DE-SC0018409, startup funding from the Department of Biochemistry and Molecular Biology, and support from AgBioResearch (MICL02454). B.H. gratefully acknowledges a generous endowment from James K. Billman, Jr., M.D. G.M. is supported by a fellowship from Michigan State University under the Training Program in Plant Biotechnology for Health and Sustainability (T32-GM110523), E.L. is supported by the NSF Graduate Research Fellowship Program (DGE-1848739), and A.B. is supported by NSF-IMPACTS Training Grant (DGE-1828149). B.H. is in part supported by the National Science Foundation under Grant Number 1737898. Any opinions, findings, and conclusions or recommendations expressed in this material are those of the author(s) and do not necessarily reflect the views of the National Science Foundation. Michigan State University occupies the ancestral, traditional, and contemporary Lands of the Anishinaabeg–Three Fires Confederacy of Ojibwe, Odawa, and Potawatomi peoples. The University resides on Land ceded in the 1819 Treaty of Saginaw.

## Author contributions

A.E.B., E.R.L., and B.H. conceived and designed the study; A.E.B. and E.R.L. performed the experiments; A.E.B. and D.M. performed and analyzed the synteny; K.H.L. assembled and annotated the genomes; B.V. and J.P.H. performed genome analyses; A.E.Y. performed ancestral state reconstruction; E.R.L. and G.P.M. analyzed the experimental data; A.E.B., E.R.L., and P.P.E. generated and analyzed the phylogenetic relationships; A.E.B., E.R.L., and B.H. wrote the manuscript; B.H. and C.R.B. supervised the project; all authors contributed to revisions.

## Competing interests

The authors declare no competing interests.
