## [Peer Review File · Nature Communications]

Uncovering a miltiradiene biosynthetic gene cluster in the Lamiaceae reveals a dynamic evolutionary trajectoryReviewers' Comments:

Reviewer #1:

Remarks to the Author:

The authors investigated the origin and subsequent evolution of a diterpenoid biosynthetic gene cluster (BGC) throughout the Lamiaceae (mint) family. They predicted a simplified version of BGC evolved in an early Lamiaceae ancestor and tried to elucidate the terpene backbones generated by the *Callicarpa americana* BGC enzymes. The work is interesting, but the writing lacks conciseness and logic. For example, Table 1 was missed in the manuscript. For the genome assembly and annotation of three novel species (*P. vulgaris*, *P. barbatus*, and *L. leonurus*), authors just gave a very brief description for the "Genome sequencing, assembly, and annotation of three Lamiaceae species" without sufficient information, e.g. the genome size estimation, the sequencing coverage, the repeat content, the number of predicted protein coding genes to judge the quality of this work. Also, the genome sizes of these three newly assembled genomes in this study were missed in Figure 2. Moreover, the experiment part such as transient expression in *N. benthamiana* needs improvement to draw the conclusion.

1. DATA AVAILABILITY : the said web page to obtain supplemental materials, genome assemblies and annotations, <https://doi.org/10.5061/dryad.w9ghx3frg>, is unavailable.
2. The Figure 1 just summarize the published works, please move it to supplementary figure.
3. Table 1 were missed from the manuscript.
4. The genome size of *L. leonurus*, *P. barbatus* and *P. vulgaris* were not provided.
5. Insufficient information was provided for the genome assembly and annotation of *L. leonurus*, *P. barbatus* and *P. vulgaris*. Additional, key information, e.g. genome size estimation, genome sequencing coverage, protein-coding gene numbers, repetitive elements, was not provided as well. Since fragmented genome is unsuitable for gene cluster study, therefore, the genome assembly quality of other published genomes should also be listed as a supplementary table for evaluating whether they were suitable for evolutionary study of biosynthetic gene cluster.
6. The missing of context information in the syntenic relationships of the miltiradiene biosynthetic gene cluster (Fig. 3) makes it is hard to evaluate the accuracy of syntenic analysis.
7. The authors manually curated the syntenic relationships, but the detail process was not described in detail.
8. The inconsistency of manuscript and Figure legend in Figs, e.g. diTPS in manuscript while TPS in figure 3 legend, 71D clade in manuscript while CYP71D clade in figure 4. In addition, how to determine *S. baicalensis* has tandem duplications of class II diTPS rather than other duplication types? How to get the conclusion of "There are also several diTPS and CYP pseudogenes presumably present from past tandem duplications"? Is there any syntenic relationships between a and b BGCs in both *S. miltiorrhiza* and *L. leonurus* ? Why "the presence of two related BGCs in both *S. miltiorrhiza* and *L. leonurus*" be the evidence of BGC has assembled and disassembled in a lineage-specific manner?
9. The logic behind "As expected, syntenic diTPSs in both subfamilies have common ancestry. Recent tandem duplications in the TPS-e and TPS-c families are evident in all examined species and contribute to lineage-specific BGC expansion" from Fig. 4 and Fig. 5, is unclear. Please give more detail explanations.
10. From Fig. 4, I think that TPS-e tree was constructed by Class I diTPS enzymes. However, it is confusing to read ", no BGC class I enzymes clustered in clade TPS-e.1.". What are IrKSL4, PbEKS, SmKSL2, and NrTPS2?
11. "All CYP76AHs in BGCs have paralogs within each cluster" is not sufficient to conclude that they were expanded by tandem duplication events. Moreover, a lot of genes between two adjacent CYP76AH in many species (Fig. 3), indicating alternative explanations. It could be tandem duplication with insertion in-between or dispersed duplication. Please consolidate the evidences.
12. Where is the *E. lutea* cluster? Is it missing from figures or manuscript?
13. The expression pattern of BGCs is a common pattern or specific in *C. americana*, please give more evidences to show that. Colors indicated the pseudogene was missing.
14. For the experimental validation, please give more evidences to show the "successfully cloned from cDNA", and "transiently expressed the genes in *Nicotiana benthamiana*".

Reviewer #2:

Remarks to the Author:

This study investigates diterpenoid biosynthetic gene evolution in the Lamiaceae as a consequence of biosynthetic gene cluster (BGC) assembly. In particular, synteny analysis has been performed for miltiradiene BGCs in the genomes of ten Lamiaceae species. The authors found conservation of core genes of the cluster and were able to reconstruct a minimal ancestral cluster. Furthermore, functional characterization was performed of the miltiradiene pathway and a new (+)-kaurene-specific pathway, both of which are part of a large BGC of *Callicarpa americana*. The presented experimental work is solid.

Many BGC of different pathways in different species and families have been described over the past decade. What sets this study apart to some extent is the comparison of a BGC across diverse lineages demonstrating species specific non-syntenic expansions of clusters beyond the core set of syntenic genes. The study exemplifies nicely the dynamics of diterpenoid BGC evolution from a simple core ancestral state. In addition, the discovery of a (+)-kaurene synthase will be beneficial for engineering novel diterpenoid biosynthetic pathways depending on this precursor.

I was a bit disappointed that the authors could not provide more specific evidence for the presence of diterpenoids in *C. americana* roots other than a characteristic fragmentation pattern, which makes it difficult to correlate the enzymatic products with what is present in the plant. How many different diterpenoids are present in *C. americana*? Could the analysis be refined?

Another aspect in which the study could be improved is the inclusion of some comparative BGC transcriptome data across the investigated Lamiaceae lineages. It would be interesting to determine to what extent syntelogs and non-syntelogs correlate with distinct tissue-specific patterns in the different species. Are expression patterns of core genes conserved in different lineages?

Reviewer #3:

Remarks to the Author:

The article by Bryson et al. describes the genomic, evolutionary, transcriptomic and biosynthetic investigation of a set of related gene clusters across genomes in the Lamiaceae family. The paper is very well-written and combines a range of different techniques to describe this intriguing case of biosynthetic conservation as well as diversification. As only relatively few complex plant biosynthetic gene clusters have been studied in detail, this paper provides valuable knowledge that widens and deepens our understanding of these important genomic loci. Particularly noteworthy is the fact that the complex biosynthetic locus in *Callicarpa americana* was shown to be composed of two 'sub-clusters' that are expressed in different tissues and encode distinct (yet related) pathways.

I have a number of queries and suggestions that may be useful to the authors in improving their paper further before publication:

- Although arguably this is a matter of taste, I personally find the introduction quite lengthy. With 1273 words, it approaches the length of a mini-review. For a research article, I would perhaps advise shortening it significantly to go more directly from the general importance of the topic to the specific challenge addressed in this paper.
- Line 171-172: 'assembled and disassembled in a lineage-specific manner'. I find this wording a bit confusing, as it almost seems to suggest that the gene clusters were independently assembled in each lineage, which is clearly not the intention.
- Line 286-287 (Figure 6): The transcriptomic comparison across tissues is a very nice feature of this manuscript. However, as explained for example here

(<https://www.ncbi.nlm.nih.gov/pmc/articles/PMC7373998/>), (log) FPKM normalization is not optimal for comparing across samples, as is done here. After all, the total number of RPKM/FPKM normalized counts for each sample may be different. Normalizing using TPM could be an alternative (although also not ideal because of compositionality); another option could be normalizing the read counts per gene as a fraction of the total set of reads mapping to the entire gene cluster as a percentage (although of course the total absolute transcript counts for the whole cluster are likely different across tissues as well). Admittedly, this will probably not change the results (much), but it may avoid unsupported conclusions being drawn from the figure.

- Line 399: 'recruitment of CamTPS10': Given that *C. americana* is very deep-branching compared to the other species studied (Figure 3), it seems to me that the data would also support loss of TPS10 in the branch leading up to the other 6 species.
- Methods: It appears that a description of the methods for transcriptome analysis (read mapping etc., Figure 6) are missing, unless Figure 6 was created based on processed data without any custom analyses. If that is the case, it would be useful to indicate that very briefly as well.
- Line 436: it would be useful to indicate here what were the criteria used during manual curation of synteny.
- Line 446: the description of how the phylogenies were constructed is confusing, as Clustal Omega is not an algorithm for phylogenetic analysis. I assume the multiple sequence alignments were generated using Clustal Omega, and the trees were generated using RAxML? Some more information would also be useful: what parameters were used for RAxML? Was the whole alignment used for phylogenetic reconstruction, or only conserved domains/positions? In general, making the alignments and trees publicly available online somewhere would be useful to readers.
- Line 449: a closing bracket seems to be missing here.
- Uploading the annotations of the three genomes to GenBank would be tremendously useful to the scientific community, as it would make it possible to e.g. find the proteins encoded in them in NCBI Blast searches. It would be really nice if authors were willing to do this and to include GenBank accession numbers of the genomes in the paper.

Bryson, Lanier et al., response to reviewer's comments

We would like to note that during the revisions we resolved the structure of one additional CTP76AH67 oxidized diterpene product, now given in Supplementary figure 11.

REVIEWER #1

The authors investigated the origin and subsequent evolution of a diterpenoid biosynthetic gene cluster (BGC) throughout the Lamiaceae (mint) family. They predicted a simplified version of BGC evolved in an early Lamiaceae ancestor and tried to elucidate the terpene backbones generated by the *Callicarpa americana* BGC enzymes. The work is interesting, but the writing lacks conciseness and logic. For example, Table 1 was missed in the manuscript. For the genome assembly and annotation of three novel species (*P. vulgaris*, *P. barbatus*, and *L. leonurus*), authors just gave a very brief description for the "Genome sequencing, assembly, and annotation of three Lamiaceae species" without sufficient information, e.g. the genome size estimation, the sequencing coverage, the repeat content, the number of predicted protein coding genes to judge the quality of this work. Also, the genome sizes of these three newly assembled genomes in this study were missed in Figure 2. Moreover, the experiment part such as transient expression in *N. benthamiana* needs improvement to draw the conclusion.

1. DATA AVAILABILITY : the said web page to obtain supplemental materials, genome assemblies and annotations, <https://doi.org/10.5061/dryad.w9ghx3frg>, is unavailable.

We confirm that the link is now working. With the addition of several new experiments, we have updated the repository. This process includes curation by the Dryad staff so the new data may not be available in the first few days after re-submission.

2. The Figure 1 just summarize the published works, please move it to supplementary figure.

Figure 1 was moved to the supplemental information.

3. Table 1 were missed from the manuscript.

Reference to Table 1 was corrected to Figure 1.

4. The genome size of *L. leonurus*, *P. barbatus* and *P. vulgaris* were not provided.

We have added the flow cytometry size estimations as well as the Supernova estimated genome size via k-mer analyses to the revised manuscript.

5. Insufficient information was provided for the genome assembly and annotation of *L. leonurus*, *P. barbatus* and *P. vulgaris*. Additional, key information, e.g. genome size estimation, genome sequencing coverage, protein-coding gene numbers, repetitive elements, was not provided as well. Since fragmented genome is unsuitable for gene cluster study, therefore, the genome assembly quality of other published genomes should also be listed as a supplementary table for evaluating whether they were suitable for evolutionary study of biosynthetic gene cluster.

We have added genome size estimation (flow cytometry and k-mer derived), genome assembly metrics from Supernova including coverage used in the assembly, gene annotation metrics including BUSCO analysis of the genome and the gene annotation, and repetitive sequence content. Where available, quality metrics of the genomes used along with references to their publications are now available in Supplementary Table 1.

6. The missing of context information in the syntenic relationships of the miltiradiene biosynthetic gene cluster (Fig. 3) makes it is hard to evaluate the accuracy of syntenic analysis.

Chromosome-level synteny figures of each species are now shown in Supplemental Fig. 2.

7. The authors manually curated the syntenic relationships, but the detail process was not described in detail.

We clarified this language to read: "Orthologs and syntenic lines were manually curated using 70% sequence identity cutoff determined by the BLASTp alignment function (Threshold = 0.05, Word Size = 3, Matrix = BLOSUM62, Gap Costs = Existence:11 Extension:1)"

8a. The inconsistency of manuscript and Figure legend in Figs, e.g. diTPS in manuscript while TPS in figure 3 legend, 71D clade in manuscript while CYP71D clade in figure 4.

We have changed the names to be consistent, using diTPS and CYP71D clade throughout.

8b. In addition, how to determine *S. baicalensis* has tandem duplications of class II diTPS rather than other duplication types? How to get the conclusion of "There are also several diTPS and CYP pseudogenes presumably present from past tandem duplications"?

We hypothesize that tandem duplication is the most likely mechanism because of their immediate phylogenetic relationships and their vicinity within the genomes. We have revised the text to clarify and removed all instances where this criterion was not met.

8c. Is there any syntenic relationships between a and b BGCs in both *S. miltiorrhiza* and *L. leonurus* ?

The syntenic relationships between species with multiple clusters can be inferred by their common syntenic relationships across species. We believe including within-species synteny lines would complicate this figure. We hope that shared syntenic lines in Fig. 2 coupled with phylogenetic evidence from Fig. 3 will lead our readers to these conclusions.

8d. Why "the presence of two related BGCs in both *S. miltiorrhiza* and *L. leonurus*" be the evidence of BGC has assembled and disassembled in a lineage-specific manner?

We have clarified the wording and hope we have addressed this concern.

9. The logic behind "As expected, syntenic diTPSs in both subfamilies have common ancestry. Recent tandem duplications in the TPS-e and TPS-c families are evident in all examined species and contribute to lineage-specific BGC expansion" from Fig. 4 and Fig. 5, is unclear. Please give more detail explanations.

We appreciate Reviewer #1 for challenging our over-simplified language. We have added specific examples of tandem duplications where appropriate and removed 'tandem' where there is no clear evidence for it.

10. From Fig. 4, I think that TPS-e tree was constructed by Class I diTPS enzymes. However, it is confusing to read ", no BGC class I enzymes clustered in clade TPS-e.1.". What are IrKSL4, PbEKS, SmKSL2, and NrTPS2?

It is true that IrKSL4, PbEKS, SmKSL2, and NrTPS2 are in the TPS-e.1 clade. Our statement claims that there were no enzymes specifically from any of the examined BGCs that phylogenetically clustered within this clade. We hope the wording changes we made help to clarify this.

11. "All CYP76Ahs in BGCs have paralogs within each cluster" is not sufficient to conclude that they were expanded by tandem duplication events. Moreover, a lot of genes between two adjacent CYP76AH in

many species (Fig. 3), indicating alternative explanations. It could be tandem duplication with insertion in-between or dispersed duplication. Please consolidate the evidences.

We have changed our language to more accurately reflect the number of species which appear to have tandem duplication of CYP76AH genes in their BGCs. We hypothesize that these genes are due to tandem duplication because in several species they appear in tandem arrays in the genomes and because they share significant sequence similarity (approximately 95% or more).

12. Where is the *E. lutea* cluster? Is it missing from figures or manuscript?

The E. lutea partial cluster did not meet our 70% sequence identity criterion, hence it was not mentioned in the main text. We have added a more in-depth figure (Supplemental Fig. 5) which will provide additional context for our conclusions. By including the E. lutea enzymes in the phylogenies (Fig. 3) we hope to present some relationships between these and the Lamiaceae enzymes without misleading our readers by including it in figures referencing synteny.

13. The expression pattern of BGCs is a common pattern or specific in *C. americana*, please give more evidences to show that. Colors indicated the pseudogene was missing.

We appreciate this suggestion and have analyzed and now added expression from each cluster using unique tissue/treatment data available on the SRA database. This can be found in Supplemental Fig. 6. We colored the pseudogenes with the color of their corresponding gene families to show how few non-related genes are present in this cluster. Language in the figure legend has been changed to clarify this.

14. For the experimental validation, please give more evidences to show the “successfully cloned from cDNA”, and “transiently expressed the genes in *Nicotiana benthamiana*”.

More information about these methods can now be found in the respective methods sections. Additionally, primers can be found in Supplemental Table 5.

REVIEWER #2

This study investigates diterpenoid biosynthetic gene evolution in the Lamiaceae as a consequence of biosynthetic gene cluster (BGC) assembly. In particular, synteny analysis has been performed for miltiradiene BGCs in the genomes of ten Lamiaceae species. The authors found conservation of core genes of the cluster and were able to reconstruct a minimal ancestral cluster. Furthermore, functional characterization was performed of the miltiradiene pathway and a new (+)-kaurene-specific pathway, both of which are part of a large BGC of *Callicarpa americana*. The presented experimental work is solid.

Many BGC of different pathways in different species and families have been described over the past decade. What sets this study apart to some extent is the comparison of a BGC across diverse lineages demonstrating species specific non-syntenic expansions of clusters beyond the core set of syntenic genes. The study exemplifies nicely the dynamics of diterpenoid BGC evolution from a simple core ancestral state. In addition, the discovery of a (+)-kaurene synthase will be beneficial for engineering novel diterpenoid biosynthetic pathways depending on this precursor.

1. I was a bit disappointed that the authors could not provide more specific evidence for the presence of diterpenoids in *C. americana* roots other than a characteristic fragmentation pattern, which makes it difficult to correlate the enzymatic products with what is present in the plant. How many different diterpenoids are present in *C. americana*? Could the analysis be refined?

We agree with this response and refined our analysis here. Extracting the entire combined root system rather than just a small portion yielded evidence for abietatriene and an oxidized abietatriene compound also made by CYP76AH68 in N. benthamiana. This is much stronger support for the biological relevance of the C. americana CYP activities we found. More highly decorated abietane diterpenoids are likely also present in the root extract. However, identifying the type and number of these would require intensive work in LCMS method refinement, purification, and NMR, and would merit a separate study beyond the scope of this work. We added the following to the main text, along with a new supplemental figure (Supplemental Fig. 14): “We analyzed a whole root extract of C. americana by GCMS and found compounds with matching retention time and mass spectra to abietatriene and the oxidized product (2c) produced by CYP76AH68. This supports the biological relevance of enzyme activities elucidated in N. benthamiana.”

2. Another aspect in which the study could be improved is the inclusion of some comparative BGC transcriptome data across the investigated Lamiaceae lineages. It would be interesting to determine to what extent syntelogs and non-syntelogs correlate with distinct tissue-specific patterns in the different species. Are expression patterns of core genes conserved in different lineages?

We appreciate this suggestion and have added expression from each cluster using unique tissue/treatment data available on the SRA database. This can be found in Supplemental Fig. 6.

REVIEWER #3

The article by Bryson et al. describes the genomic, evolutionary, transcriptomic and biosynthetic investigation of a set of related gene clusters across genomes in the Lamiaceae family. The paper is very well-written and combines a range of different techniques to describe this intriguing case of biosynthetic conservation as well as diversification. As only relatively few complex plant biosynthetic gene clusters have been studied in detail, this paper provides valuable knowledge that widens and deepens our understanding of these important genomic loci. Particularly noteworthy is the fact that the complex biosynthetic locus in *Callicarpa americana* was shown to be composed of two ‘sub-clusters’ that are expressed in different tissues and encode distinct (yet related) pathways.

I have a number of queries and suggestions that may be useful to the authors in improving their paper further before publication:

1. Although arguably this is a matter of taste, I personally find the introduction quite lengthy. With 1273 words, it approaches the length of a mini-review. For a research article, I would perhaps advise shortening it significantly to go more directly from the general importance of the topic to the specific challenge addressed in this paper.

We agree with Reviewer #3 that our introduction is extensive, however since we cover a variety of fields of study (i.e., biochemistry, genomics, genetics) we hope to provide a broad audience with sufficient background. We have made adjustments of language and sentence structure to address the reviewer’s concerns and moved figure 1 to the supplementary information.

2. Line 171-172: ‘assembled and disassembled in a lineage-specific manner’. I find this wording a bit confusing, as it almost seems to suggest that the gene clusters were independently assembled in each lineage, which is clearly not the intention.

We have changed this language to read: "It is evident that each BGC, while maintaining the core miltiradiene genes, has undergone some lineage-specific independent evolution." We hope to convey that while the core genes had been assembled in a common ancestor, each species had evolved uniquely along the way.

3. Line 286-287 (Figure 6): The transcriptomic comparison across tissues is a very nice feature of this manuscript. However, as explained for example here (<https://www.ncbi.nlm.nih.gov/pmc/articles/PMC7373998/>), (log) FPKM normalization is not optimal for comparing across samples, as is done here. After all, the total number of RPKM/FPKM normalized counts for each sample may be different. Normalizing using TPM could be an alternative (although also not ideal because of compositionality); another option could be normalizing the read counts per gene as a fraction of the total set of reads mapping to the entire gene cluster as a percentage (although of course the total absolute transcript counts for the whole cluster are likely different across tissues as well). Admittedly, this will probably not change the results (much), but it may avoid unsupported conclusions being drawn from the figure.

We appreciate Reviewer #3's insight into presenting expression data. We have now included a figure using TPM in Supplemental Fig. 6. After research and direct comparison, we have chosen to keep the expression data presented in Fig. 3 measured in FPKM for several reasons. Firstly, this data is not intended to be a statistical comparison but rather a general overview of the BGC validity and activity in the whole plant. The broad tissue types and developmental stages presented here cannot be assumed to have comparable baseline RNA content, as is necessary for direct comparison using TPM. Secondly, analysis using TPM instead of FPKM de-emphasizes some of the expression patterns, which are consistent with our metabolite extractions showing the presence of relevant compounds and we validated expression of the genes by cloning from the corresponding RNA. We hope that providing now both types of analysis of this data will show a more complete story.

4. Line 399: 'recruitment of CamTPS10': Given that *C. americana* is very deep-branching compared to the other species studied (Figure 3), it seems to me that the data would also support loss of TPS10 in the branch leading up to the other 6 species.

*We agree with the comment that presence of CamTPS10 solely in *Callicarpa americana* could support loss of this gene in a common ancestor outside *Callicarpoideae* instead of recruitment within *Callicarpoideae*. However, CamTPS10 makes a novel backbone, suggesting it is unique to *C. americana*. It is also phylogenetically distinct from all the class I enzymes present in the BGCs, so it is less likely to have been a duplication and neofunctionalization event in this BGC. Although not conclusive, we also found no evidence of a CamTPS10-like gene in our outgroup BGC. Since we would like to present a minimal ancestral cluster, we chose to present a single class I gene based on these evidences. We have added qualifying language to this sentence in the manuscript to clarify these scenarios.*

5. Methods: It appears that a description of the methods for transcriptome analysis (read mapping etc., Figure 6) are missing, unless Figure 6 was created based on processed data without any custom analyses. If that is the case, it would be useful to indicate that very briefly as well.

We have added methods for transcriptome analysis in Fig. 3 and Supplemental Fig. 6.

6. Line 436: it would be useful to indicate here what were the criteria used during manual curation of synteny.

We clarified this language to read: “Orthologs and syntenic lines were manually curated using 70% sequence identity cutoff determined by the BLASTp alignment function (Threshold = 0.05, Word Size = 3, Matrix = BLOSUM62, Gap Costs = Existence:11, Extension:1)”

7. Line 446: the description of how the phylogenies were constructed is confusing, as Clustal Omega is not an algorithm for phylogenetic analysis. I assume the multiple sequence alignments were generated using Clustal Omega, and the trees were generated using RAxML? Some more information would also be useful: what parameters were used for RAxML? Was the whole alignment used for phylogenetic reconstruction, or only conserved domains/positions? In general, making the alignments and trees publicly available online somewhere would be useful to readers.

We have now clarified how ClustalOmega and RAxML were used. We added the model and algorithm used by RAxML. We included which parts of the sequences were used in the phylogenetic analysis. Our alignments for each phylogeny are now available in our supplemental files on Dryad (<https://doi.org/10.5061/dryad.w9ghx3frq>).

8. Line 449: a closing bracket seems to be missing here.

We corrected this.

9. Uploading the annotations of the three genomes to GenBank would be tremendously useful to the scientific community, as it would make it possible to e.g. find the proteins encoded in them in NCBI Blast searches. It would be really nice if authors were willing to do this and to include GenBank accession numbers of the genomes in the paper.

Raw genomic reads have been deposited to the SRA database; Sequences of the enzymes characterized from C. americana have been deposited to GenBank. Genome assemblies and complete annotations have been deposited to our Dryad (<https://doi.org/10.5061/dryad.w9ghx3frq>). We plan to upload our genome assemblies and annotations to NCBI after re-submission. By providing the data in Dryad as well, we can make the data available sooner.

Reviewers' Comments:

Reviewer #1:

Remarks to the Author:

All issues and questions raised have been addressed well by the authors.

Minor comments:

Line 488: "Plants were returned to the controlled growth room 76 °C", please carefully check the temperature.

Reviewer #2:

Remarks to the Author:

I have reviewed the revised manuscript. The authors included new experimental and analytical results in response to my comments and I have no further concerns at this time.

Reviewer #3:

Remarks to the Author:

I am happy to see the revised version of the manuscript, which adequately addressess all the points I previously raised, as well as those of the other referees.

It would be great if the authors could finalize submission of the annotated genomes to NCBI before final publication, so that the accession numbers can be included in the Data Availability statement.

Point-by-point response to the reviewers' comments, Bryson, Lanier and co-workers, NCOMMS-22-13476B

NCBI submission of the three novel genomes was completed and hyperlinks for the live files are embedded in the manuscript file. All author checklist points were answered.